

# High-frequency productivity estimates for a lake from free-water CO$_2$ concentration measurements

Maria Provenzale[1], Anne Ojala[2,3], Jouni Heiskanen[4], Kukka-Maaria Erkkilä[1], Ivan Mammarella[1], Pertti Hari[3], and Timo Vesala[1,3]

[1]Department of Physics, University of Helsinki, Helsinki, Finland
[2]Department of Environmental Sciences, University of Helsinki, Lahti, Finland
[3]Department of Forest Sciences, University of Helsinki, Helsinki, Finland
[4]ICOS ERIC Head Office, Helsinki, Finland

*Correspondence to:* Maria Provenzale (maria.provenzale@helsinki.fi)

**Abstract.** Lakes are important actors in biogeochemical cycles and a powerful natural source of CO$_2$. However, they are not yet fully integrated in carbon budgets, and the carbon cycle in the water is still poorly understood. In freshwater ecosystems, productivity studies have usually been carried out with traditional methods (bottle incubations, $^{14}$C technique), which are imprecise and have a poor temporal resolution. Consequently, our ability to quantify and predict the net ecosystem productivity

($NEP$) is limited: the estimates are prone to errors and the $NEP$ cannot be parameterized from environmental variables. Here we expand the testing of a free-water method based on the direct measurement of the CO$_2$ concentration in the water. The approach was proposed already in 2008, but was tested on a very short data set (3 days) under specific conditions (autumn turnover); despite showing promising results, it has not been used ever since. We tested the method under different conditions (summer stratification, typical summer conditions for boreal dark-water lakes) and on a much longer data set (40 days), and

quantitatively validated it comparing our data and productivity models. We were able to evaluate the $NEP$ with a high temporal resolution (minutes) and found an excellent agreement with the models. We also estimated the parameters of the productivity-irradiance (PI) curves that allow the calculation of the $NEP$ from irradiance and water temperature. Overall, our work shows that the approach is suitable for productivity studies under a wider range of conditions, and is an important step towards developing it so that it becomes even more general.

# 1 Introduction

Lakes are very important actors in the local and global carbon cycles (Battin et al., 2009; Tranvik et al., 2009). They both fix carbon, through the photosynthesis of the in-lake primary producers, and release it, through the respiration of all the aquatic organisms (primary producers, consumers and microbes), through photochemical reactions and by transmitting the received carbon from the catchment (lateral transport) back to the atmosphere in gaseous form (CO$_2$). Many lakes - especially the olig-

otrophic ones typical of high latitudes - are net heterotrophic systems where the rate of community respiration exceeds that of primary production (Cole et al., 1994; Sobek et al., 2003); this contributes to make lakes one of the most important natural sources of greenhouse gases (Raymond et al., 2013). However, they are not yet fully integrated in the local and global carbon





budgets, and the lacustrine carbon cycle is still poorly known (Cole et al., 2007).

In freshwater ecology, productivity studies have usually relied on the light and dark bottle method (Gaarder and Gran, 1927) and the $^{14}$C labeling technique (Steemann Nielsen, 1951; Peterson, 1980; Bender et al., 1987; Søndergaard, 2000). The first provides estimates both of the gross and the net primary productivity, whereas the latter gives an estimate that is between the

gross and the net productivity, depending on the incubation time. These traditional methods require time- and effort-demanding measurements and have a poor temporal resolution. Periods of high productivity are easily missed (Karl et al., 2003) and, because of the low temporal resolution, the non-linear relationship between photosynthetically active solar radiation ($PAR$) and photosynthesis cannot be properly investigated. As a consequence, carbon balances may be imprecise and for instance the net ecosystem productivity ($NEP$) cannot be parameterized robustly as a function of ambient variables. Moreover, communities

enclosed in bottles experience light and nutrient conditions far from the natural ones, since the movement of water or of the organisms themselves is limited (Mallin and Paerl, 1992; Reynolds, 2006), and the results can be unrealistic. Thus, advances in the methodology are necessary, to better estimate freshwater ecosystems productivity and to expand our understanding of the carbon cycle in the water column.

In the last 15 years, free-water methods, not requiring sampling and incubation, have become more common. These methods,

however, are usually based on the measurement of the $O_2$ concentration in the water, which is then used as a proxy for $CO_2$ (Hoellein et al., 2013; Solomon et al., 2013): this introduces uncertainties (Staehr et al., 2010). The respiratory quotient that has to be applied when transforming rates from $O_2$ to $CO_2$ has, in fact, large variations (Berggren et al., 2012).

To study the in-water photosynthesis and respiration, Hari et al. (2008) proposed a free-water method based on the direct measurement of the $CO_2$ concentration in the water with non-dispersive infra-red (NDIR) $CO_2$ probes, associated with a con-

comitant assessment of the $CO_2$ flux between the lake and the atmosphere. The probes are designed to measure the $CO_2$ concentration in the air, but by building a gas collection system the concentration in the water is obtained. Similar probes have been used also in Johnson et al. (2010), albeit not for productivity studies. The temporal resolution is of five seconds, more than a hundredfold improvement over the traditional approaches. A requirement of the method is the concomitant assessment of the $CO_2$ flux from the lake to the atmosphere. Information on the in-lake vertical $CO_2$ flux is also needed (and, ideally, on

the lateral transport as well). If such data are missing the method can be applied under specific conditions (e.g. stable stratification); it still allows the parameterization of the $NEP$ from $PAR$ and water temperature, from which the $NEP$ can then be calculated under different conditions.

In Hari et al. (2008), the method was tested on a small boreal lake in Finland over three days only, during the autumn turnover. A cross-comparison was carried out between different measurement methods, but the $NEP$ was not mathematically parameter-

ized and the method was not quantitatively verified. Despite the very short data set and the specific conditions, the results were promising: the relationship between $PAR$ and $NEP$ was clearly visible, the measured respiration rate was 16 times higher than with the bottle method and the measured productivity was 5 times higher than with the $^{14}$C technique. The numbers are in line with previous studies: Pace and Prairie (2005) reported similar discrepancies between an oxygen-based free-water approach and the bottle method in small lakes in Michigan, and a tendency of the $^{14}$C method to underestimate the productivity is

well known (Howarth and Michaels, 2000). However, the method has been overlooked and has not been used for productivity





calculations since 2008, possibly because of the limited testing.

Here we tested the method of Hari et al. (2008) on a different boreal lake, under different conditions and on a much longer data set, quantitatively verifying it. We continuously collected data for four summers, and then we focused on the periods when the lake was stably stratified, i.e. summer conditions typical of boreal dark-water lakes, in order to rule out the lateral $CO_2$

flux and the $CO_2$ flux from the deeper layers of the lake. Overall, we analysed 40 days of data. We calculated the $NEP$ using the equations that are typically used in forest ecology, where high-frequency measurements are more common, in an effort of harmonizing the procedures between different fields. Once we had the $NEP$ with a high temporal resolution, we verified the relationship between the $NEP$ and irradiance, using a saturating Michaelis-Menten model. We found an excellent agreement between the data and the model. From that, we could also estimate the parameters of the productivity-irradiance (PI) curves,

specific to the in-lake communities. These parameters are very important because they allow the calculation of the $NEP$ from $PAR$ and water temperature.

Whilst our efforts were mainly focused on method testing and development, we also checked whether the parameters of the PI curves we estimated changed significantly between the years. Our goal was to gather information on how sensitive the parameters are to variations in the communities living in the lake or in the environmental conditions. We investigated whether their

behaviour could be related to their main drivers, water temperature and irradiance.

## 2  Materials and procedures

### 2.1  Study site

The study site is the boreal lake Kuivajärvi, in southern Finland (61°50.743' N, 24°17.134' E). Lake Kuivajärvi is typical dark-water boreal lake. It is small and oblong, and it is surrounded by managed coniferous forests. Its surface area is 0.62 km$^2$

and its length is 2.6 km; its mean depth and maximum depth are 6.3 m and 13.2 m respectively. The lake is humic (surface median DOC concentration = 11.8 mg l$^{-1}$ in 2011) and mesotrophic (surface median annual total nitrogen concentration = 370 $\mu$g l$^{-1}$ and annual total phosphorus concentration = 14 $\mu$g l$^{-1}$ in 2011), with a chlorophyll $a$ concentration in the surface layer usually between 3 and 5 $\mu$g l$^{-1}$ (median 4.8 $\mu$g l$^{-1}$ in 2011), with summer values that can reach 30 $\mu$g l$^{-1}$ (Miettinen et al., 2015). The lake is dark coloured: the Secchi depth ranges from 1.2 to 1.5 m (Heiskanen et al., 2015). The lake is dimictic and

it is frozen for five months every year on average; the spring turnover occurs immediately after the ice out in late April or early May, and after the turnover a thermocline starts developing. The thermocline deepens until the autumn turnover, and finally the lake freezes over in late November or early December (Heiskanen et al., 2015; Mammarella et al., 2015). A map with the location and bathymetry of the lake is available in the supplemental information (Figg. S1 and S2).

### 2.2  Measurements

All the instruments were mounted on a raft, which was moored in the middle of the lake (see the supplemental information, Fig. S2, for the exact position of the raft on the lake). To measure the $CO_2$ concentration in the water, a closed system consisting of





a NDIR probe (CARBOCAP® GMP343, Vaisala Oyj, Vantaa, Finland) for the $CO_2$ concentration in the air, gas impermeable tubes (stainless steel and teflon) and a submerged gas permeable tube (silicone rubber, Rotilabo 9572.1, Carl Roth GmbH and Co. KG, Karlsruhe, Germany) was built; the air was circulated continuously in the system by a diaphragm pump (KNF Neuberger Micro gas pump, KNF Neuberger AB, Stockholm, Sweden). Analog voltage outputs were used, logged with a Nokeval

RMD680 serial transmitter to a ASCII-file on a Windows-based computer. Since silicone rubber has an excellent permeability to $CO_2$ (Carignan, 1998; Hari et al., 2008), the concentration of $CO_2$ in the air circulating in the system equilibrated with that in the water around the submerged tube. Hence, the $CO_2$ concentration in the water could be obtained from that in the air using the dependence of $CO_2$ solubility on temperature and pressure. The $CO_2$ concentration in the water $C_{CO_2}$ (dissolved $CO_2$), in $\mu\text{mol m}^{-3}$, was calculated as

$$C_{CO_2} = \chi_{CO_2} P K_H, \tag{1}$$

where $\chi_{CO_2}$ is the $CO_2$ gas phase mole fraction in the tube measured by the probe (in $\mu\text{mol mol}^{-1}$), $P$ is the total air pressure inside the system and $K_H$ is Henry's law constant (temperature dependent). For more details on the setup see Hari et al. (2008), Heiskanen et al. (2014) and the supplemental information (Fig. S3). The $CO_2$ concentration in the water was measured at a depth of 0.2 m (determined by the depth of the submerged silicone tube). The system was operating continuously from May to

September 2010-2014, but the data from year 2012 are not used here due to technical problems. The silicone tube was cleaned once a week to avoid biofouling and changed once a month. The $CO_2$ sensors were calibrated using span and zero gases. A thermistor chain of 16 Pt100 resistance thermometers (depths: 0.2, 0.5, 1.0, 1.5, 2.0, 2.5, 3.0, 3.5, 4.0, 4.5, 5.0, 6.0, 7.0, 8.0, 10.0, and 12.0 m) was deployed and a $PAR$ sensor (LI-192, LI-COR Inc., Nebraska, USA) for photosynthetic photon flux density ($PPFD$) was submerged in the water at the same depth as the $CO_2$ measurement (0.2 m). An eddy covariance (EC)

system (with ultrasonic anemometer USA-1, Metek GmbH, Germany, and closed-path infra-red gas analyzer LI-7000, LI-COR Inc., Nebraska, USA, replaced in 2011 by enclosed-path infra-red gas analyzer LI-7200, LI-COR Inc., Nebraska, USA) was used to detect the $CO_2$ flux between the lake and the atmosphere. The fluxes were calculated and quality screened according to the standard procedures, see Vesala et al. (2006), Mammarella et al. (2009), Mammarella et al. (2015) and the supplemental information (Sect. S2). All the instruments were powered by mains electricity.

**2.3   Calculation of the net ecosystem productivity**

The net ecosystem productivity ($NEP$, $\mu\text{mol}(CO_2)\,\text{m}^{-2}\,\text{s}^{-1}$), also called net ecosystem uptake, can be defined as

$$NEP = GPP - R_h, \tag{2}$$

where $GPP$ (gross primary productivity) is the amount of carbon fixed by the primary producers through photosynthesis and $R_h$ (ecosystem respiration) is the amount of carbon lost through respiration, both autotrophic and heterotrophic. The $NEP$ is

the opposite of the net ecosystem exchange ($NEE$), whose expression can be derived from the conservation of mass. Hence, considering the mass balance of $CO_2$ in the mixed layer of the lake, where most of the photosynthesis takes place, and assuming



that lateral transport of $CO_2$ is of no importance, the $NEP$ can also be expressed as

$$NEP = -NEE = -\int_{-h_{\mathrm{mix}}}^{0} \frac{\partial C_{\mathrm{CO_2}}(z,t)}{\partial t}dz - F_{\mathrm{a}} + F_{\mathrm{u}}; \tag{3}$$

see Fig. 1 for a schematic representation. In Eq. (3), $C_{\mathrm{CO_2}}$ is the $CO_2$ concentration in the water calculated from Eq. (1), $F_{\mathrm{a}}$ is

the $CO_2$ flux between the lake and the atmosphere (positive if from the lake to the atmosphere), $F_{\mathrm{u}}$ is the $CO_2$ flux between the

deeper and the mixed layer of the lake (positive if upwards), $t$ is time and $z$ is depth. The integration is computed between the

mixing depth $h_{\mathrm{mix}}$ and the surface. The mixing depth is defined as the depth at which the water temperature starts decreasing

faster than one degree per meter (Staehr et al., 2010); in our study the average value was $h_{\mathrm{mix}} = 1.5$ m. Given the dark water

colour and the resulting low light conditions in the lake, there was no benthic primary production in the profundal zone. While

$C_{\mathrm{CO_2}}$ was measured by the probe and $F_{\mathrm{a}}$ by the EC system, we had no precise way of measuring $F_{\mathrm{u}}$. This is the main reason

behind our choice to limit the analysis to the summer days when the lake was stably stratified and it was safe to assume no gas

was exchanged through the thermocline: $F_{\mathrm{u}} = 0$. The periods of stable stratifications were chosen on the basis of temperature

plots and of the Schmidt stability of the lake, calculated with the LakeAnalyzer program, according to Read et al. (2011). For

all the chosen days, the stability ($Sc$) is $> 100\,\mathrm{Jm^{-2}}$. However, not all days with $Sc > 100\,\mathrm{Jm^{-2}}$ were used: days with strong

winds or stable atmospheric stratification were discarded because of their impact on fluxes (for more detailed information, see

the end of this section). For the time series of isotherms for the whole summers (from 1 June to 31 August), and the time series

of isotherms, Schmidt stability, $CO_2$ concentration and $PAR$ at 0.2 m and air temperature for the periods of stable stratification

chosen for analysis each year see the supplemental information (Figg. S4-S14). Overall, we analyzed 40 days in 10 periods

occurring between mid-June and the end of July of each year.

It is worth pointing out that Eq. (3) resembles the equation used in terrestrial ecology to estimate the $NEP$. In fact, e.g. in

forest EC calculations (Foken et al., 2012), neglecting lateral transport, the $NEP$ is

$$NEP = -NEE = -\int_{0}^{h_{\mathrm{m}}} \rho_{\mathrm{d}} \frac{\partial \chi_{\mathrm{CO_2}}(z,t)}{\partial t}dz - \rho_{\mathrm{d}} \overline{w'\chi'_{\mathrm{CO_2}}}. \tag{4}$$

In Eq. (4), $\rho_{\mathrm{d}}\,\chi_{\mathrm{CO_2}}$ ($\rho_{\mathrm{d}}$ = dry air density, $\chi_{\mathrm{CO_2}}$ = $CO_2$ mixing ratio) replaces $C_{\mathrm{CO_2}}$ as the $CO_2$ concentration in the air instead

of in the water, and $z$ is the height (with $h_{\mathrm{m}}$ = measuring height); $\rho_{\mathrm{d}} \overline{w'\chi'_{CO_2}}$ is $F_{\mathrm{a}}$, the $CO_2$ flux from the forest to the

atmosphere, calculated as the covariance between the fluctuations of the vertical wind velocity and the gas mixing ratio. From

the analogy, we can see that using high-frequency $CO_2$ concentration measurements and Eq. (3) in aquatic ecology reduces the

gap with terrestrial ecology, where high-frequency measurements are common, and leads towards a greater uniformity in the

equations and procedures.

Resuming our calculation of the $NEP$ in aquatic ecosystems through Eq. (3), to increase the precision of the concentration

data, half-hourly averages of $C_{\mathrm{CO_2}}$ from the raw 5-sec data were used. A 30-min resolution is enough to capture the variations

caused by the biological activity and at the same time filter out the ones caused by the physical mixing of the water (Staehr

et al., 2010). However, the EC data set, which also has a resolution of 30 minutes, had many gaps, due to inherent problems




of the EC technique (wind not blowing along the lake, stability or not fully developed turbulence resulting in quality criteria not met) and technical problems (instrument failures). Approximately 70% of the data points for the summers were rejected or missing, with occurrences of consecutive days having no acceptable data points at all. Hence, for our data set, a point by point calculation of $F_\mathrm{a}$ in (3) was not possible. Even though in general it would not be needed, we had to use a daytime and
a nighttime average value for $F_\mathrm{a}$; we mainteined the half-hourly calculation of the $NEP$ to preserve the temporal resolution. The daytime and nighttime average $F_\mathrm{a}$ values were calculated separately for each year, combining all the studied periods of water stable stratification of the same summer. Before doing so, we checked that the environmental conditions (temperature and relative humidity cycles, incoming radiation, wind speed and direction, atmospheric stability) were similar for all the analysed days in the summer. In particular, since wind and atmospheric stability have the greatest influence on the fluxes (given that the
lake water is thermally stratified), as verified in Heiskanen et al. (2014), we discarded any day with winds $> 5$ m s$^{-1}$ or with stable atmospheric stratification. For the remaining days, the wind was always weak, with averages $< 2.5$ m s$^{-1}$; at such low speeds, the influence of the wind on the flux is negligible (Cole and Caraco, 1998). Under these circumstances (i.e. warm and sunny summer days without strong wind events), the $CO_2$ flux is expected to have similar daily cycles across the analysed days. Day and night were defined on the basis of $PAR$. When using $PAR$, we are referring to the average $PAR$ value in the mixed
layer, obtained from the 0.2 m value through the lake light extinction coefficient (1.5). The threshold between day and night was set to $20\,\mu\mathrm{mol(ph)\,m^{-2}\,s^{-1}}$ and it was chosen by calculating the average value of $PAR$ at which the $CO_2$ concentration in the water started decreasing in the morning after accumulating during the night, or increasing again in the evening. Using this procedure, "day" represents the fraction of the time series when photosynthesis dominates over respiration, and not the times when photosynthesis takes place in absolute terms.
At this point, we were able to calculate the half-hourly values of $NEP$ for each period.

## 2.4 Relationship between $NEP$ and $PAR$

In humic lakes, the photosynthesis is strongly driven by $PAR$, and the relationship can be described for instance by the Michaelis-Menten equation (Caperon, 1967; Kiefer and Mitchell, 1983). Assuming that the daytime respiration rate equals the nighttime respiration rate and that they depend exponentially on temperature (Carignan et al., 2000), the $NEP$ can be
expressed as

$$NEP = GPP - R_\mathrm{h} = \frac{p_\mathrm{max}\,PAR}{PAR+b} - r_0\,Q_{10}^{T/10}. \tag{5}$$

In (5), $T$ is the water temperature (in °C) and $Q_{10}$ is a non-dimensional temperature coefficient whose generally accepted value for freshwater communities is 2 (in the literature, values between 1.88 and 2.19 are reported: Reynolds (1984); Raven and Geider (1988); Davison (1991)); $p_\mathrm{max}$, $b$ and $r_0$ represent the maximum potential photosynthetic rate, the half-saturation
constant (i.e. the value of $PAR$ at which the photosynthetic rate is half of the maximum rate) and the basal respiration rate, respectively. These parameters are important, since they allow the calculation of $NEP$ from water temperature and $PAR$. After calculating the $NEP$, we plotted the $NEP$ versus irradiance curves. We then fitted the $NEP$ data to the model (Eq. (5))





with the least-squares fitting method, in order to check the agreement between the data and the model and in order to estimate $p_{\max}$, $b$ and $r_0$.

## 3   Assessment

The $NEP$ had the same trend as the incoming radiation, as expected; it had bigger negative values during the night, when only

respiration took place, and smaller negative values during the day, when photosynthesis contributed with an uptake of $CO_2$. Anyhow, the net productivity values are almost always negative, meaning that the ecosystem, overall, is heterotrophic and a source of $CO_2$. Figure 2 shows the $CO_2$ concentration change in time over the mixed layer (the first term in Eq. (3)), which is usually referred to as storage flux in forest ecology calculations, the $NEP$, the average daytime and nighttime values of the $CO_2$ flux ($F_{a_{\text{day}}}$ and $F_{a_{\text{night}}}$) and $PAR$ for a sample period of stable stratification in July 2010, representative of the analyzed

periods. The nine-day period in Fig. 2 is the longest of the entire data set. Generally, stable stratification lasted from two to five days; its short duration is due to the oblong shape of the lake, that makes it sensitive to wind action: as soon as the wind increases the mixing is enhanced (although complete mixing takes place only in spring and autumn). Figure 3 displays the $NEP$ versus $PAR$. We decided to draw a different plot for each year, instead of combining all the data points from all the years, since the conditions ($PAR$ and water $T$) varied from year to year. The figure also displays the model curve, calculated

using the average water $T$ of the studied periods of each year. From the plots, we can see that for low values of $PAR$ the $NEP$ was strongly negative; then, as $PAR$ increased, the $NEP$ quickly increased as well; however, as already noted, the $NEP$ always remained negative, indicating net heterotrophy. None of the years exhibited signs of photoinhibition: the $NEP$ did not seem to decrease even at high ($> 500\,\mu\mathrm{mol(ph)}\,\mathrm{m}^{-2}\,\mathrm{s}^{-1}$) values of $PAR$. In case of photoinhibition, the addition of a specific term accounting for it in (5) would have been needed, but here it was not necessary. Differences can be seen between the years,

with 2014 showcasing the smallest values of $NEP$. Year 2014 was particularly hot, so the strongly negative $NEP$ can be due to increased respiration rates; year 2010 though displays the highest values of $NEP$ despite having an intermediate average water temperature. Figure 4 shows the same plots, but focuses on the dependence of the $NEP$ on $T$. The model is calculated for different values of water $T$, ranging from the minimum to the maximum water temperatures recorded during the studied periods of each year. The $NEP$ decreases with increasing temperature, due to higher respiration rates. Finally, Fig. 5 features

3D plots of the data and the curves, to visualize simultaneously the dependence of the $NEP$ on $PAR$ and water $T$. The curves in Figg. 2-5 have the expected trends, and this confirms that the data and the equation used are proper tools for estimating the $NEP$ at a high temporal resolution. The results of the fittings of the $NEP$ versus $PAR$ and $T$ are reported in Table 1 below. Considering the assumptions we had to adopt, there is a very good agreement between the model and the data: the $R^2$ values range from 0.71 to 0.84. This clearly indicates that the method used here allows the $NEP$ to be accurately parameterized as a

function of irradiance and water temperature.

An in-depth analysis of the values of the model parameters (reported in Table 1) and their inter-annual variability is beyond the scope of this paper. However, some comments are possible. The maximum photosynthetic rate $p_{\max}$ ranged between 1.55 (2014) and 0.63 (2013) $\mu\mathrm{mol(CO_2)}\,\mathrm{m}^{-2}\,\mathrm{s}^{-1}$, and it was higher in 2011 and 2014 than in 2010 and 2013. The half-saturation





constant $b$ ranged between 22 (2010) and 33 (2013) $\mu\mathrm{mol(ph)\,m^{-2}\,s^{-1}}$, being higher in 2011, 2013 and 2014 than in 2010. The values of $b$ are relatively small. It indicates that the phytoplankton communities were well adapted to the low light conditions (boreal area and dark-water lake) and were able to start photosynthesising even when the incoming radiation was small. The basal respiration $r_0$ ranged between 0.228 (2010) and 0.482 (2014) $\mu\mathrm{mol(CO_2)\,m^{-2}\,s^{-1}}$, being higher in 2011 and 2014 than

in 2010 and 2013, as was the case with $p_{\max}$. The parameters however do not appear to be strictly correlated to each other, and a clear and uniform pattern in their behavior cannot be identified.

We checked whether the differences in the parameter values between the years are statistically significant. This also gives an indication whether the parameters should be re-assessed each year or can be considered lake-specific. We calculated the parameters difference and its confidence interval (calculated as the uncertainty of the difference, from the confidence intervals

of the parameters themselves), and verified whether it overlapped 0. The value of $b$ does not change significantly between any of the years: this means that the algal communities adapted to the light conditions in a similar way every year. The values of the other parameters change: $p_{\max}$ is comparable only between 2011 and 2014, and $r_0$ is never comparable. The difference in $p_{\max}$ and $r_0$ can be due to different total algal biomass in the lake. In general, we can say that there are statistically significant differences between the years. Variations in the environmental conditions might have led to changes in the communities living

in the lake, or the communities might have responded differently to the environmental conditions; $p_{\max}$ and $r_0$ seem to be more sensitive to variations than $b$.

Finally, we investigated whether the changes of the model parameters can be explained in terms of changes, during the analyzed periods, of the ambient variables that act as $NEP$ drivers: water temperature and irradiance. The model parameters and the average, minimum and maximum values of water $T$ and $PAR$ for each year are reported in Table 2 (only the 40 analyzed

days are considered in these statistics). In 2010 and 2011 the surface water temperature had similar average values, 22.9 and 22.7 °C respectively. Year 2013 was slightly colder, with an average value of 21.5 °C, while year 2014 was warmer, with an average value of 25.6 °C. The minimum temperatures of the study periods were similar for 2010 and 2013 ($\approx 20$ °C), slightly higher for 2011 (20.7 °C) and notably higher for 2014 (23.2 °C). The maximum temperatures ranged between 23.5 (2013) and 28.3 (2014) °C. Overall, 2013 can be considered as a cold year, 2014 as a hot year, 2010 and 2011 as intermediate years. The

temperature variation pattern between the years cannot be easily linked to the variations in $b$. Concerning $p_{\max}$, even though the largest value of $p_{\max}$ is associated with the warmest year (2014), and the smallest value of $p_{\max}$ with the coldest year (2013), years 2010 and 2011 had different values of $p_{\max}$ despite having similar temperatures. Besides, $p_{\max}$ and $b$ are expected to depend more strongly on $PAR$ than on $T$. Conversely, $r_0$ can be expected to be larger when temperatures are higher. This happened in 2011 and 2014, but not in 2010, which still had relatively high temperatures. Possible explanations are changes

in the $Q_{10}$ value, or the influence of other environmental variables. We did not investigate further possible changes in the $Q_{10}$ value, both because we did not have an independent way to estimate it, and because its range is narrow according to the literature (Reynolds, 1984; Raven and Geider, 1988; Davison, 1991). Concerning $PAR$, in the analyzed periods the average values in the mixed layer ranged from 162 (2013) to 227 (2014) $\mu\mathrm{mol(ph)\,m^{-2}\,s^{-1}}$, being higher in 2010 and 2011 than in 2013, and notably higher in 2014 than in all the other years. Remarkably also in 2014, despite the high values of $PAR$, the

communities did not show signs of photoinhibition (a $PAR_{\max}$ value of 741 for the mixed layer corresponds to a surface value



of $\approx 1900\,\mu\mathrm{mol(ph)\,m^{-2}\,s^{-1}}$, given the light extinction coefficient of the lake of 1.5). Higher average $PAR$ values could be responsible for larger $p_{\max}$ values, as observed in 2011 and 2014, and partially in 2010. However, the average $PAR$ values are very similar in 2010 and 2011, while $p_{\max}$ values are not. Still, the very low value of $p_{\max}$ in 2013 could be explained by the low $PAR_{\mathrm{ave}}$ value. The variations of $b$ between the years, though, cannot be linked to the changes in $PAR$: 2013 and

2014, despite having very different $PAR$ values, had similar $b$ values. The trend in $r_0$ also cannot be associated with the trend in $PAR$ between the years. From what said so far, the changes of $PAR$ and water temperature cannot fully account for the changes in the model parameters. A more extensive analysis would require more information on the algal communities living in the lake each year, and as already stated is beyond the scope of this paper.

## 4    Discussion

Firstly, it is important to notice that we are working under the assumption that the $NEE$, which is what can be measured, is equal in magnitude to the $NEP$. This concept is widely accepted in the scientific community (Aubinet et al., 2012), for forests as well as for other environments such as lakes.

The lateral transport of $CO_2$ had to be ruled for the sake of the calculations; we are of course fully aware of the lake being a 3D dynamic system, and we hope in the future to further develop the method we used so that it will work under more general

conditions. Besides, since this study focuses on the summer periods when the lake was stably stratified and there were no high winds or rains, the lateral transport is not expected to play a significant role here. A similar challenge is encountered in forest ecology studies as well, where the lateral transport in the air (advection) is also usually neglected.

The daytime and nighttime average values of the $CO_2$ flux were always positive, albeit having lower values during the day than during the night. This is not surprising: many lakes, especially at high latitudes, are supersaturated with respect to $CO_2$;

as a result, the $CO_2$ flux is from the lake to the atmosphere also during the day, when the aquatic primary producers are photosynthesising and absorbing $CO_2$. The $NEP$ values where in fact also always negative, confirming heterotrophy. This is in agreement with (Cole et al., 1994; Sobek et al., 2003).

Regarding oligotrophic lakes, it has been suggested that diurnal patterns in the epilimnion stratification and water convective motions (causing nighttime upwelling of $CO_2$) are important drivers of the diurnal variation of the surface water $CO_2$ concentra-

tion (Åberg et al., 2009). Lake Kuivajärvi though is mesotrophic (chl $a$ 5-30 $\mu$gl$^{-1}$ during summer) and the primary production can be assumed to be the main driver of the $CO_2$ concentration, as observed also in some other lakes with high chl $a$ (Hanson et al., 2003; Huotari et al., 2009). Also, we implemented strict selection criteria of the analyzed periods to minimize the effect of upwelling $CO_2$: the thermistor data indicate that the winds, despite being weak, were strong enough to keep the top 1.5 m of the water column well mixed both day and night, without however disrupting the thermocline. Thus, no sign of hypolimnetic

upwelling was detected. Under these conditions, diurnal stratification patterns and convective motions had a minor impact on the mixed layer of our lake. It is also important to note that the photochemical production of $CO_2$ is generally negligible in humic lakes (Jonsson et al., 2001); its maximum contribution to the flux for a lake with similar characteristics as the one in our study lake was $< 4\%$ over the whole growing season, and was detectable only in the top 10 cm of the water column (Vähätalo





et al., 2000; Ojala et al., 2011).

In aquatic sciences, other models for describing the dependence of photosynthesis on irradiance are more commonly used than the Michaelis-Menten equation. The Michaelis-Menten equation was chosen in an effort of harmonizing productivity studies between aquatic and forest sciences, in order to study the carbon cycle consistently in the forest-lake continuum. However, we

checked whether other models provided a better fit to the data. We used the equations by Smith (1936) and by Jassby and Platt (1976). Even though they agreed well with the data, they did not perform significantly better than the Michaelis-Menten equation: the $R^2$ and $RMSE$ values of the fits were very similar. Hence, we decided to proceed with our first choice. The Smith (1936) and Jassby and Platt (1976) model equations and fit statistics are reported in the supplementary information (Sect. S3 and Table S1).

In this study, we could not clearly link the environmental variables to the changes in the Michaelis-Menten model parameters, and more information on the algal communities living in the lake would have been required in order to extend the analysis. However, it is important for us to stress that the simplicity of this method lies in the fact that to estimate the parameters, which can then be used to calculate the productivity for each year, such information is not needed. Besides, an extensive application of the method would also allow for a comparison of the parameters between different lakes and different times, and

the understanding of the relationship between the parameters and the environmental conditions or the specific phytoplankton communities would improve.

Our analysis was hindered by the problematic EC data set: due to inherent EC limitations and technical problems, the data set had many gaps and average flux values had to be used. The calculations could be improved with a better EC data set or chamber measurements performed regularly.

For further development, it would also be good to have measurements for $F_\mathrm{u}$, the $CO_2$ flux from the deeper layer to the surface layer of the lake, in order not to limit the analysis to isothermal or stable stratification conditions. For example, water column turbulence measurements could be added to the $CO_2$ concentration and temperature measurements.

## 5   Conclusions

The high-frequency direct $CO_2$ concentration measurement method suggested in Hari et al. (2008) and tested only on 3 days

of data under autumn turnover conditions was tested more extensively here, on a dataset of 40 days and under stable stratification conditions typical of summer for dark-water lakes. The method proved to be suitable for lake productivity studies under isothermal (Hari et al., 2008) or stable stratification conditions: its high temporal resolution allowed us to calculate the net ecosystem productivity at a temporal scale of minutes. A quantitative comparison between the calculated $NEP$ and the modeled $NEP$ was also carried out for the first time, and it showed a very good agreement between the two, further validating

the method. From that, we were able to accurately parameterize the net productivity as a function of the ambient variables, estimating the productivity parameters typical of the communities in the lake.

Overall, we believe that the method proposed in Hari et al. (2008) and further tested, verified and developed here, with the explicit parameterization of the $NEP$ from $PAR$ and water $T$, represents a great improvement over the traditional approaches




(bottle method and $^{14}$C technique). At the present stage it is still very system specific, and assumptions about lateral and vertical $CO_2$ exchange and photo-oxidation had to be made (negligible lateral exchange and photo-oxidation, no in-lake vertical exchange). Still, our study is an important step towards testing and developing the approach so that it becomes more general, also given the scarcity or even lack of high-frequency direct $CO_2$ measurements for productivity studies. We are looking for

further contributions by the research community and we think the method should be widely adopted, in order first to gather more information about its usability under different conditions and then also to have a broader network of productivity studies on lakes. This is all the more true given that it is also easy to set up and relatively inexpensive. Its only requirement is a concomitant estimation of the $CO_2$ flux from the lake to the atmosphere. In our case the EC technique was used, which is expensive and can be laborious in the data processing phase. However, chamber measurements for example are an equally good option.

Additionally, the method also relies on equations that are typically adopted in terrestrial ecology studies for the calculation of the $NEP$, where high-frequency measurements are more commonplace than in aquatic research. Extensively applying the method would reduce the gap in the $CO_2$ exchange measurements between aquatic and terrestrial ecology, which is beneficial in the framework of integrating research in different ecosystems, for which purpose a common language between different disciplines is needed. It would also help us achieve a better understanding of the biological processes behind the $CO_2$ exchange.

This, in turn, would expand our knowledge on the carbon cycle in the water, which is still limited, and would lead to a better integration of aquatic ecosystems in the local and global carbon budgets.

*Code and data availability.*   The data sets and the codes used in this paper can be obtained from the authors upon request.

*Competing interests.*   The authors declare that they have no conflict of interest.

*Acknowledgements.*   This study was funded by the University of Helsinki and the Finnish Cultural Foundation - Häme fund (Hämeen ra-
hasto - Suomen Kulttuurirahasto). Support also came from the Academy of Finland, through the Academy Professor projects (1284701 and 1282842), the CarLAC project (281196), ICOS-Finland (1281255) and the Finnish center of Excellence in Atmospheric sciences, and from the EU project GHG-LAKE (612642).





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



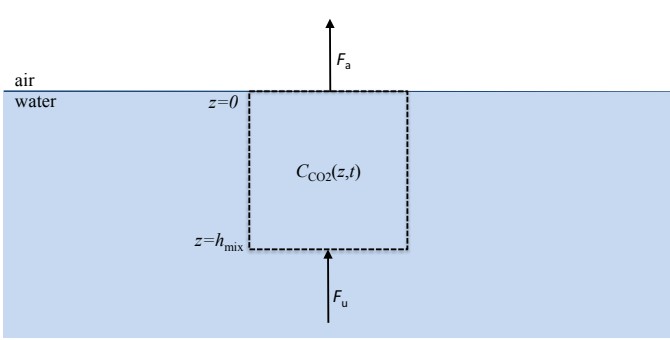

**Figure 1.** A schematic representation of the mass balance in the mixed layer of the lake. $C_{CO_2}$ is the $CO_2$ concentration in the water, $F_a$ is the $CO_2$ flux between the lake and the atmosphere (positive if from the lake to the atmosphere), $F_u$ is the $CO_2$ flux from the deeper to the mixed layer of the lake (positive if upwards), $t$ is time, $z$ is depth and $h_{mix}$ is the mixed layer depth.




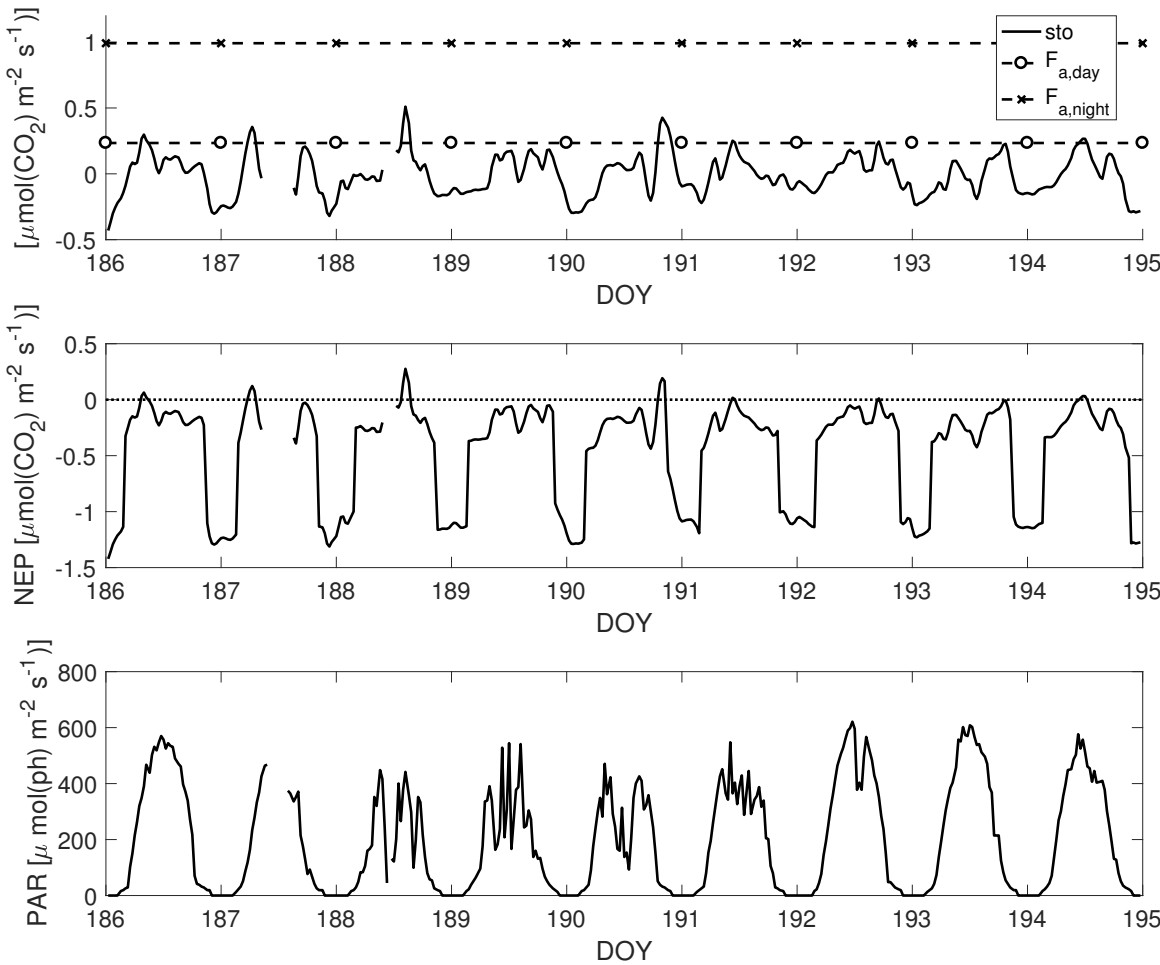

**Figure 2.** A sample period of stable stratification in July 2010, representative of the studied periods. In the upper panel the solid line ($sto$) is the first term of Eq. (3), the $CO_2$ concentration change in time over the mixed layer, which is usually referred to as storage flux in forest ecology calculations; the dashed horizontal lines are the daytime and nighttime average $CO_2$ fluxes from the lake to the atmosphere ($F_a$). In the central panel, the solid line is the $NEP$ and the dotted line is the zero rate. In the lower panel the solid line is the $PAR$ (photon flux density measured in the $PAR$ wavelength range) average value in the mixed layer.





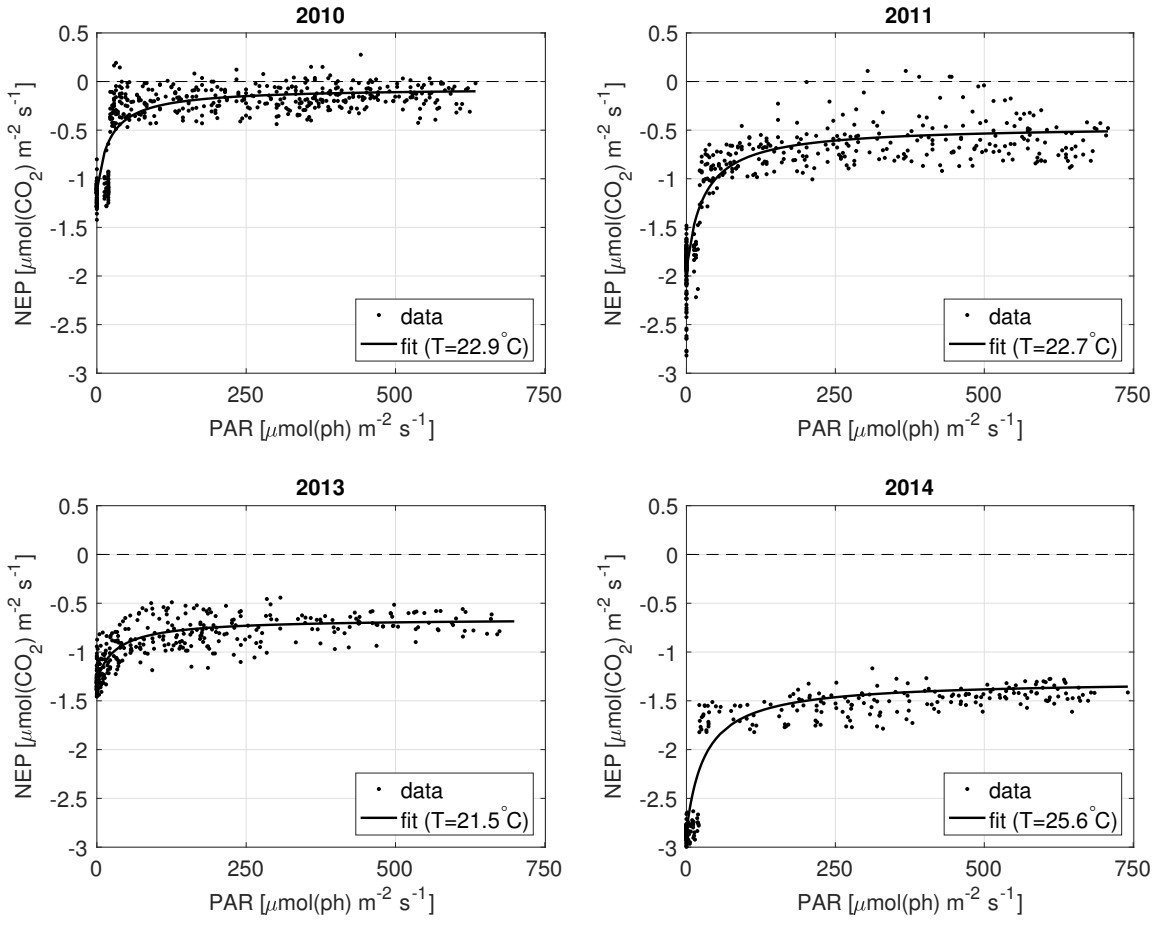

**Figure 3.** The $NEP$ versus $PAR$ plots for each year; the fitted curve shown is calculated using the average water $T$ of the studied periods of the year.





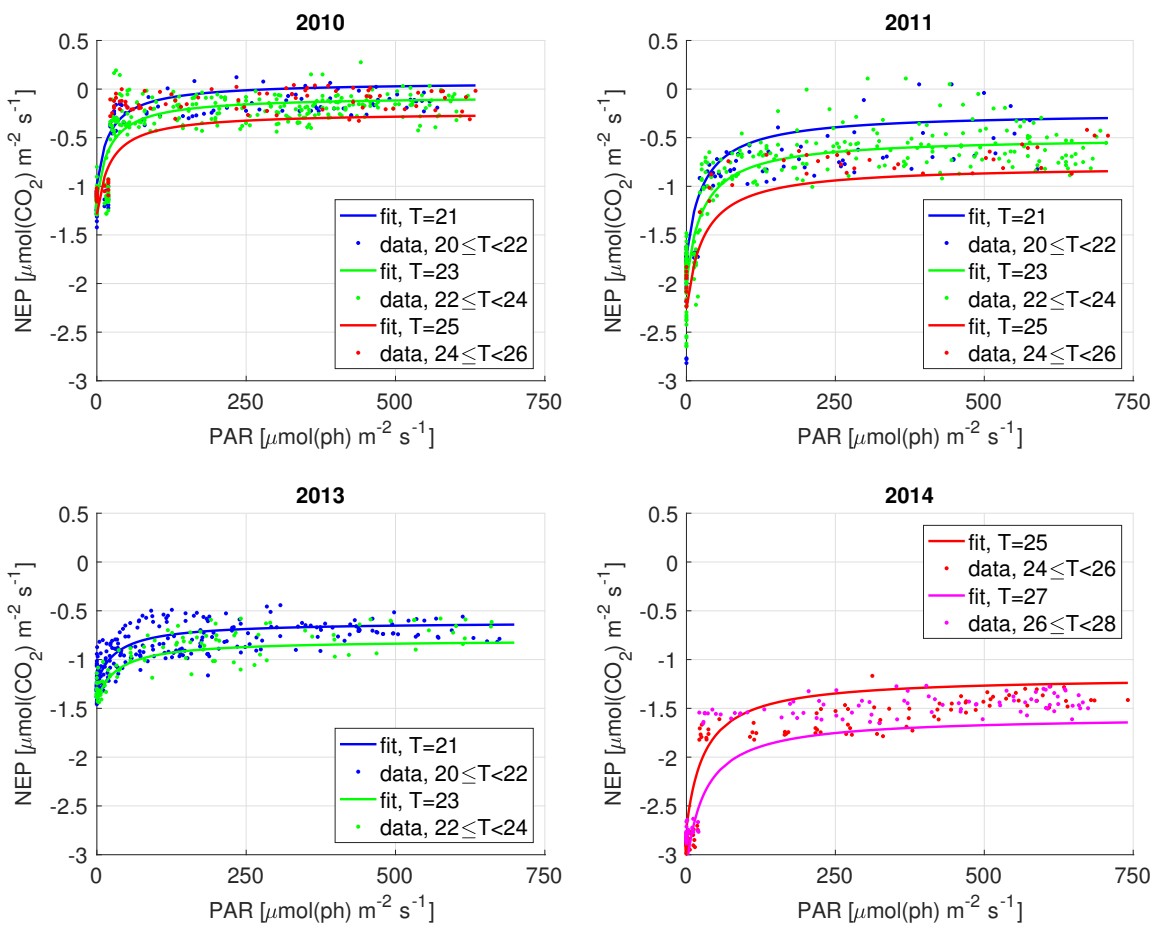

**Figure 4.** The $NEP$ versus $PAR$ plots for each year. Data points are color-classified according to water temperature classes, and the curves are calculated for the different temperatures; note that the curves are not individual fits, but are the result of the year's 3D fit, evaluated for the different temperatures. Water $T$ is in $^\circ$C.



**Figure 5.** Data and fitted $NEP$ versus $PAR$ and water $T$ 3D-curves for each year.



**Table 1.** Parameters of the $NEP$ vs $PAR$ and water $T$ model (Eq. (5)), with 95% confidence intervals, and fit statistics.

| Year | $p_{\max}$ $\left[\mu\mathrm{mol(CO_2)\,m^{-2}\,s^{-1}}\right]$ | $b$ $\left[\mu\mathrm{mol(ph)\,m^{-2}\,s^{-1}}\right]$ | $r_0$ $\left[\mu\mathrm{mol(CO_2)\,m^{-2}\,s^{-1}}\right]$ | $R^2$ | $RMSE$ $\left[\mu\mathrm{mol(CO_2)\,m^{-2}\,s^{-1}}\right]$ |
|---|---|---|---|---|---|
| 2010 | $1.05 \pm 0.05$ | $22 \pm 5$ | $0.228 \pm 0.008$ | 0.73 | 0.23 |
| 2011 | $1.47 \pm 0.06$ | $29 \pm 6$ | $0.399 \pm 0.009$ | 0.84 | 0.25 |
| 2013 | $0.63 \pm 0.04$ | $33 \pm 10$ | $0.290 \pm 0.007$ | 0.71 | 0.14 |
| 2014 | $1.55 \pm 0.10$ | $31 \pm 11$ | $0.482 \pm 0.013$ | 0.74 | 0.33 |





**Table 2.** The model parameters as in Table 1 ($p_{\max}$ and $r_0$ in $\mu\mathrm{mol(CO_2)\,m^{-2}\,s^{-1}}$, $b$ in $\mu\mathrm{mol(ph)\,m^{-2}\,s^{-1}}$), average, minimum and maximum values of water $T$ (°C) and $PAR$ in the mixed layer ($\mu\mathrm{mol(ph)\,m^{-2}\,s^{-1}}$) for the studied periods of each year.

| Year | $p_{\max}$ | $b$ | $r_0$ | $T_{\mathrm{ave}}$ | $T_{\min}$ | $T_{\max}$ | $PAR_{\mathrm{ave}}$ | $PAR_{\max}$ |
|------|------|----|-------|------|------|------|------|------|
| 2010 | 1.05 | 22 | 0.228 | 22.9 | 19.9 | 26.2 | 195 | 634 |
| 2011 | 1.47 | 29 | 0.399 | 22.7 | 20.7 | 25.3 | 197 | 708 |
| 2013 | 0.63 | 33 | 0.290 | 21.5 | 20.0 | 23.5 | 162 | 699 |
| 2014 | 1.55 | 31 | 0.482 | 25.6 | 23.2 | 28.3 | 227 | 741 |