# Peer review of "High-frequency productivity estimates for a lake from free-water $CO_2$ concentration measurements"

_Biogeosciences, 2017_

## Referee Comment (RC1) · J. Zwart (Referee) · 4 Dec 2017

Provenzale and co-authors describe a study in which they use high-frequency CO2 measurements to estimate lake net ecosystem production (NEP) and describe variation in NEP using water temperature and light in the upper mixed layer. The authors constrained their study to well-stratified periods so as to satisfy assumptions of CO2 transport in the lake. Michaelis-Menten model fits to the estimated NEP were very good and parameters varied between years, for reasons which the authors speculate (e.g. algal community composition). This study advances CO2 metabolism research in lakes; however, I think the results and discussion sections need a lot of restructuring

to effectively describe the important results of this study. I make several suggestions below that I think will improve the manuscript.

Hanson et al. (2003, Limnology & Oceanography 48: 1112-1119) also estimated GPP, R, and NEP using CO2 measurements for 2-4 days in each of their study lakes, so this isn't the first time CO2 measurements have been used for metabolism studies other than Hari et al. (2008).

Do the authors have evidence that lateral transport of CO2 negligible in their study lake? Are there stream or groundwater inflows to the lake that may transport CO2? Even in lakes that are fairly isolated from the surrounding landscape, lateral transport of CO2 can be significant. (see Vachon et al. 2016 doi: 10.1002/lno.10454), and some autotrophic lakes can have significant CO2 outgassing, indicating a decoupling from NEP and CO2 dynamics (Bogard and del Giorgio 2016 doi: 10.1002/2016GB005463). More details on lake hydrologic characteristics and the influence of lateral transport or lack thereof would be good. What is the lake water residence time? Are there significant inlet streams? Etc.. I also think it would be useful to indicate when or for what type of lakes that this method might be useful. This is the second lake that has tested this method, but do the authors think that this method can be applied to every lake? If not, then why not?

Page 4 and 5: NEP is the net biological conversion of organic carbon to inorganic carbon while NEE is equal to NEP + inorganic sinks/sources of CO2. So I think it is incorrect to state that negative NEE is the same as NEP on page 4 line30. See Lovett et al. (2006 doi: 10.1007/S10021-005-0036-3) for an in-depth discussion of terminology and Stets et al. (2009, doi: 10.1029/2008JG000783) for an application to lakes. I think it is correct to say that NEP = -NEE if lateral transport of CO2 (and other inorganic sinks / sources) are negligible, so equation 3 seems correct, but only under this assumption.

I don't see how this manuscript harmonizes terrestrial vs. aquatic studies other than using a similar term (M-M dynamics as harmonizing isn't very convincing). And the

authors don't give any good reasons for why harmonizing terrestrial and aquatic C cycling research is needed. Sprinkling in forest ecology references here and there (e.g. page 7 line 8, Figure 2 legend) seems like a cheap connection to make to terrestrial systems. I think this paper would be stronger if the authors did not try to compare to terrestrial systems and instead focused on the merits of using $CO_2$ in addition to or in place of $O_2$ to estimate metabolism in aquatic systems (e.g. respiratory quotient different than 1, etc. . .).

Page 2 line 29: What do you mean "the NEP was not mathematically parameterized" in the Hari et al. (2008) analysis? Does this mean that NEP was not explained by PI curves?

Page 5 line7: It is unclear if hmix was set at 1.5m for the entire study period (due to stable stratification and setting Fu to zero) or if this is calculated at the frequency of the temperature measurements. If it is calculated at a high-frequency interval, do the authors account for vertical entrainment of hypo $CO_2$ into epi when thermocline deepens and epi $CO_2$ into hypo when thermocline shallows?

Page 5 line 19: is "e.g." supposed to be NEP?

Page 5 lines 19-27: I don't think equation 4 and the paragraph surrounding it adds very much to the MS and is distracting to the methods section. It is also unclear what the 'gap with terrestrial ecology' is and how using $CO_2$ measurements reduces this undefined gap. Was using different methods of ecosystem productivity really creating a separation between terrestrial and aquatic studies?

Page 6 lines 1-4: Fa was not possible at 30 min resolution; did you ever compare to a model of gas flux and fill in data gaps that way? i.e. why not use Heiskanen et al. (2014) gas flux model?

Page 6 line 28: I'm assuming $Q_{10}$ is set to 2, but the authors should be explicit to make this clear.

Page 7: The results section is not very well organized and is a combination of results that do seem to fit in the same paragraph. For example, the first paragraph covers results from Figure 2-5 and table 1 and does not flow well together. Break these up into individual paragraphs with topic sentences so that the reader knows what point the authors are trying to make with the paragraph.

Page 7 line 6: do not use colloquial language such as "Anyhow, ..."

Page 7 line 12-13: remove "Figure 3 displays the NEP versus PAR."

Page 7 lines 18-19: get rid of "In case of .... it was not necessary." This doesn't add anything to the fact that there was no photoinhibition.

Page 7 line 20-21: why is respiration more negative with hotter years? Be explicit. Because Rh is more temperature dependent than GPP?

Page 7 line 26: change Figg. to Figure

Page 7 line 31-32: Get rid of "An in-depth analysis... some comments are possible." Since you do spend three paragraphs of the results discussing these parameters. Replace with a more constructive topic sentence.

Page 8 line 7-10. Move to methods rather than results

Page 8 line 13-14: get rid of "In general, we can say that there are statistically significant differences between the years."

Page 9 line11: but see Lovett et al. 2006 where NEP does not equal –NEE.

Page 9 line 21-22: "This is in agreement with..." who? The citations in parenthesis? And what is in agreement with them? That lakes are heterotrophic when NEP is negative? Or that many lakes are heterotrophic?

Page 10 line 10-16: This is a confusing paragraph. The authors state that 1) more info on algal communities was needed to explain differences in fitted parameters, but 2) this

isn't needed because the whole point of this method is to be simple and parsimonious, but 3) this method should be applied in many lakes to make links between parameters and environmental conditions / algal communities. I don't know what point the authors are trying to make with this paragraph.

Page 10 line 28-29: How is there a comparison between the calculated NEP and modeled NEP when you fit the model to the calculated NEP? To make a statement like this, it seems like you should be training the model on a set of the calculated NEP data and then verifying with a separate set of the calculated NEP data. Also, what do you mean calculated NEP vs. modeled NEP was compared for first time? Compared for the first time using the MM method? I know there are many other examples where predictor variables are used to model lake NEP, so the authors will have to be more specific here.

Page 10 lines 32-33: this is not a clear sentence.

I don't think figure 1 is necessary.

Figure 2-5: each dot represents a day or a 30 min interval? Please specify in legend

Can tables 1 & 2 be combined? It would just add 5 more columns.

Signed: Jacob Zwart

---

## Referee Comment (RC2) · Anonymous Referee #2 · 19 Dec 2017

The manuscript "High-frequency productivity estimates for a lake from free-water CO2 concentration measurements" presents a method to assess NEP in aquatic ecosystems. While interesting, the method requires an independent measurement of (1) the flux of CO2 between the atmosphere and the water surface, (2) usage of high-frequency in situ CO2 sensors, and - at least in the present approach – (3) conditions under which lateral advection fluxes (both within the water column and in the atmosphere) are limited. Overall, the methodological approach and science appear sound, but of limited utility due to known issues with eddy covariance and chamber methods for determining water to atmosphere fluxes of CO2. That is, the atmospheric turbulence required for eddy flux determination is often not present at night, while the stratification

required for the water column (i.e. to satisfy the assumption of no lateral fluxes in the lake) would be violated under higher windspeeds. Thus, the overall measurements are constrained to a methodological "sweet spot". The authors do not explore the potential limitations of only making measurements during these ideal conditions, which were identified for 10 summer days over a number of years for a lake in Finland. While the results for the measured days are interesting, there should be some effort to describe what these NEP rates represent relative to the other seasons (early open water after thaw, etc.), as well as efforts towards uncertainty estimates of the NEP rates and how this uncertainty cascades into the least-squares regressions for modeled parameters.

The authors present in essence a case study of implementation of a method presented by Hari et al (2008), with the suggestion that the method has been overlooked and has not been used for NEP because of limited testing (P 2 L35 – P3 L1). This logic seems a bit circular, and misses the point that there are few eddy covariance studies over aquatic systems. The authors suggest that determination of the atmospheric flux of $CO_2$ could be made with chambers rather than by eddy flux, but do not discuss limitations of chamber measurements, which are not insignificant. Some discussion on how chamber measurements and in situ measurements could be co-located would be useful. As well, the study makes the assumption that $[CO_2]$ is uniform in the mixed layer, but this assumption does not appear to have been tested for confirmation.

It seems problematic that the authors calculate NEP based on time varying dC/dt, but use mean values for daytime and nighttime fluxes of the atmospheric flux (essentially static values). Perhaps there is additional information in the eddy flux data that could be used to propagate uncertainty in NEP calculations? For example, the standard deviation of the $F_a$ term for each day could be useful. The authors state (P6 L13) that the $CO_2$ flux is expected to have similar daily cycles across the analysed days, but it is not clear that the magnitude of the fluxes should be similar across days. What is the basis for this assumption?

Specific comments P1 L11: Here, the model fit is described as "excellent", while later

it is described as "very good" on P7 L28. Providing some metrics that would qualify as excellent should be included in the abstract.

P1 L19: change to "...in gaseous form (primarily as CO2)."

P4 L5: What is the permeability of silicone to CO2 relative to the diffusion rate of CO2 in water at the temperatures experienced in this study?

P6 L29: It would be helpful to present more information describing how pmax, b and ro are determined. Which equations were used to solved for these three unknowns?

P7 L26: "The curves in Figg. 2-5 have the expected trends, and this confirms that..." – this sounds like confirmation bias.

P7 L29: "This clearly indicates that the method used here allows the NEP to be accurately parameterized as a function of irradiance and water temperature." What seems to be missing here is uncertainty assessment on NEP. If NEP is not well constrained (since it is calculated from Eqn 3 assuming static rates for the daytime and nighttime CO2 fluxes between the lake and the atmosphere), how can the model fits be characterized without consideration of the uncertainty in the "measured" NEP vs. the modelled fit?

P8 L10: "The value of b does not change significantly between any of the years" – but Table 1 and 2 show it to vary by 50% between years. This seems rather significant. The later statement that pmax and ro are more sensitive to variation seems to be a statement that wasn't formally tested through sensitivity analysis.

P9 L14: "We hope in the future to further develop the method" – this kind of statement doesn't belong in a Discussion section.

Compare P9 L6-7: "the changes of PAR and water temperature cannot fully account for the changes in the model parameters" with P7 L29 "the method used here allows the NEP to be accurately parameterized as a function of irradiance and water temperature."

[Figure]

P11 L7: I would not describe eddy covariance over a water surface as "relatively inexpensive".

P17 Fig 3 for 2014: What explains the large separation between NEP values for low values of PAR and the jump in NEP just as PAR increases a bit?

---

## Author Response (AR1)

**Author comments (ACs) to the referee comments (RCs) on the manuscript "High-frequency productivity estimates for a lake from free-water CO₂ concentration measurements", by Provenzale et al. (bg-2017-412)**

The authors would like to thank the referees for the time they invested in reading and assessing the manuscript, and for the comments they provided. We believe that the feedback from the referees has allowed us to substantially improve our work. The authors would also like to thank the editor for the time and consideration.

In the following file, the blue colour denotes our answers to the reviewers, the orange colour denotes the changes we made to the manuscript.

Attached after our answers is the marked-up version of the manuscript.

**RC1**

Provenzale and co-authors describe a study in which they use high-frequency $CO_2$ measurements to estimate lake net ecosystem production (NEP) and describe variation in NEP using water temperature and light in the upper mixed layer. The authors constrained their study to well-stratified periods so as to satisfy assumptions of $CO_2$ transport in the lake. Michaelis-Menten model fits to the estimated NEP were very good and parameters varied between years, for reasons which the authors speculate (e.g. algal community composition). This study advances $CO_2$ metabolism research in lakes; however, I think the results and discussion sections need a lot of restructuring to effectively describe the important results of this study. I make several suggestions below that I think will improve the manuscript.

We thank the referee for the suggestions. We followed them to restructure the Results and Discussion sections.

Hanson et al. (2003, Limnology & Oceanography 48: 1112-1119) also estimated GPP, R, and NEP using $CO_2$ measurements for 2-4 days in each of their study lakes, so this isn't the first time $CO_2$ measurements have been used for metabolism studies other than Hari et al. (2008).

We added this information to the manuscript.

The manuscript now reads: "our study is an important step towards testing and developing the approach so that it becomes more general, also given the scarcity or even lack of high-frequency direct $CO_2$ measurements for productivity studies (we are aware of only one other study where free-water $CO_2$ measurements were used for metabolism studies (Hanson et al., 2003).", P12 L22-25 in the new version of the manuscript, P13 L26-28 in the marked-up version.

Do the authors have evidence that lateral transport of $CO_2$ negligible in their study lake? Are there stream or groundwater inflows to the lake that may transport $CO_2$? Even in lakes that are fairly isolated from the surrounding landscape, lateral transport of $CO_2$ can be significant. (see Vachon et al. 2016 doi: 10.1002/lno.10454), and some autotrophic lakes can have significant $CO_2$ outgassing, indicating a decoupling from NEP and $CO_2$ dynamics (Bogard and del Giorgio 2016 doi: 10.1002/2016GB005463). More details on lake hydrologic characteristics and the influence of lateral transport or lack thereof would be good. What is the lake water residence time? Are there significant inlet streams? Etc..

We added the hydrologic characteristics of the lake.

The manuscript now reads: "Most of the inflow is through a permanent stream in the northern end, while the role of groundwater is small during summer. Temporary inflows appear at snowmelt, through several small ephemeral streams. The outflow is located at the southern end. The residence time was 522 days in 2011 and 655 days in 2013.", P3 L28-31 in both versions of the manuscript.

As for the lateral transport, Dinsmore et al. (2013) studied the $CO_2$ concentration discharge in six sites, including lake Kuivajärvi. They concluded that most of the $CO_2$ discharge for this site happens at snowmelt or during strong rain events in the autumn. The assumption that lateral transport of $CO_2$ is, under our conditions, negligible is also confirmed by the fact that the $CO_2$ concentrations in the mixed layer only exhibit a diurnal cycle, and no long-term trend is observable (see the Figures in the supplemental information).

We added this information in the manuscript, which now reads: "The lateral transport of $CO_2$ had to be ruled for the sake of the calculations. A similar challenge is encountered in forest ecology studies as well, where the lateral transport in the air (advection) is also usually neglected. We are of course fully aware of the lake being a 3D dynamic system. Besides, since this study focuses on the summer periods when the lake was stably stratified and there were no high winds or rains, the lateral transport is not expected to play a significant role here. This assumption is supported by Dinsmore et al. (2013), who showed that for lake Kuivajärvi most of the $CO_2$ discharge happens at snowmelt or during heavy rains in the autumn. It is also supported by the mixed layer $CO_2$ concentration time series, which show no sign of a long-term trend on top of the diurnal cycles (see Figg. S5-S14 in the supplemental information).", P10 L25-31 in the new version of the manuscript, P11 L8-21 in the marked-up version.

I also think it would be useful to indicate when or for what type of lakes that this method might be useful. This is the second lake that has tested this method, but do the authors think that this method can be applied to every lake? If not, then why not?
In principle, we think that the method could be applied to any lake under any conditions, with an expansion of the instrumental set-up.

We added this information to the manuscript, which now reads: "At the current stage, the method we present here is still very system specific, and assumptions about lateral and vertical $CO_2$ exchange and photo-oxidation had to be made (negligible lateral exchange and photo-oxidation, no in-lake vertical exchange). However, the method can in principle be applied to any lake and under any condition, with an expansion of the instrumental set-up. Measurements or estimates of $F_u$, the $CO_2$ flux from the deeper layer to the surface layer of the lake, would be needed in order not to limit the analysis to isothermal (as in Hari et al., (2008)) or stable stratification (as here) conditions. This could be achieved for example adding water column turbulence measurements to the $CO_2$ concentration and temperature measurements. Chemical measurements would be needed to apply the method in clear-water lakes, where photo-oxidation could play an important role. Finally, information about $CO_2$ discharge would be needed for lakes or periods when lateral transport is not negligible.", P11 L34-P12 L8 in the new version of the manuscript, P12 L33-P13 L8 in the marked-up version.

Page 4 and 5: NEP is the net biological conversion of organic carbon to inorganic carbon while NEE is equal to NEP + inorganic sinks/sources of $CO_2$. So I think it is incorrect to state that negative NEE is the same as NEP on page 4 line 30. See Lovett et al. (2006 doi: 10.1007/S10021-005-0036-3) for an in-depth discussion of terminology and Stets et al. (2009, doi: 10.1029/2008JG000783) for an application to lakes. I think it is correct to say that NEP = -NEE if lateral transport of $CO_2$ (and other inorganic sinks/sources) are negligible, so equation 3 seems correct, but only under this assumption.
We agree.

The manuscript now reads: "Provided that there are no inorganic sinks or sources of $CO_2$, the NEP is the opposite of the net ecosystem exchange (NEE)", P4 L32-P5 L1 in both versions of the manuscript.

I don't see how this manuscript harmonizes terrestrial vs. aquatic studies other than using a similar term (M-M dynamics as harmonizing isn't very convincing). And the authors don't give any good reasons for why harmonizing terrestrial and aquatic C cycling research is needed. Sprinkling in forest ecology references here and there (e.g. page 7 line 8, Figure 2 legend) seems like a cheap connection to make to terrestrial systems. I think this paper would be stronger if the authors did not try to compare to terrestrial systems and instead focused on the merits of using $CO_2$ in addition to or in place of $O_2$ to estimate metabolism in aquatic systems (e.g. respiratory quotient different than 1, etc. . .).

We agree that we did not clearly state what we meant. Our effort was to harmonize the procedures that are used to calculate productivity from measurements in different ecosystems (and not specifically M-M dynamics, which we used here as a validation for our calculated NEP, together with other models (Smith (1936) and Jassby and Platt (1976), see the supplemental information)).

We stated our intentions more clearly. Also see the answer to the "Page 5 lines 19-27" comment.

Page 2 line 29: What do you mean "the NEP was not mathematically parameterized" in the Hari et al. (2008) analysis? Does this mean that NEP was not explained by PI curves?
A PI curve is reported in Hari et al. (2008) Fig. 4, with the data points and a modelled NEP curve. However, the mathematical expression of the modelled NEP curve is not provided, and neither is information on its agreement with the data.

Page 5 line 7: It is unclear if $h_{mix}$ was set at 1.5m for the entire study period (due to stable stratification and setting $F_u$ to zero) or if this is calculated at the frequency of the temperature measurements. If it is calculated at a high-frequency interval, do the authors account for vertical entrainment of hypo $CO_2$ into epi when thermocline deepens and epi $CO_2$ into hypo when thermocline shallows?
$h_{mix}$ was set to 1.5 m for the entire study period.

The manuscript now reads: "the average value for the entire study period was $h_{mix} = 1.5$ m", P5 L10 in the new version of the manuscript, P5 L10-11 in the marked-up version.

Page 5 line 19: is "e.g." supposed to be NEP?
"e.g." stood for "for example", we agree that it was unclear.

The manuscript now reads: "considering for example forest EC calculations", P5 L24-25 in the new version of the manuscript, P5 L25 in the marked-up version.

Page 5 lines 19-27: I don't think equation 4 and the paragraph surrounding it adds very much to the MS and is distracting to the methods section. It is also unclear what the 'gap with terrestrial ecology' is and how using $CO_2$ measurements reduces this undefined gap. Was using different methods of ecosystem productivity really creating a separation between terrestrial and aquatic studies?
We believe that the idea of finding a common language between different fields (aquatic and terrestrial productivity studies) is important. Ongoing European projects and infrastructures such as ICOS and RINGO for example have tasks related to this harmonization need. For these reasons, we decided to keep the equation and the paragraph. However, we now explain in a hopefully clearer way what the gap is and why in our opinion it is important to harmonize the methods.

The manuscript now reads: "High-frequency measurements for productivity are common in forest ecology. They are, however, less common in aquatic ecology, where traditional approaches are still widespread despite their limitations (low temporal resolution, unnatural conditions). Having different methodologies and different time resolutions creates a gap between the two fields, and

makes comparing the estimates more difficult. Given that the terrestrial and aquatic ecosystems are a continuum through which carbon is cycled, using shared procedures is a step in the direction of connecting and integrating these ecosystems, in order to have more precise carbon budgets and a deeper knowledge of the carbon cycle.", P5 L29-P6 L4 in the new version of the manuscript, P5 L30-P6 L6 in the marked-up version.

Page 6 lines 1-4: $F_a$ was not possible at 30 min resolution; did you ever compare to a model of gas flux and fill in data gaps that way? i.e. why not use Heiskanen et al. (2014) gas flux model?
We compared the available EC data with the model from Heiskanen et al. (2014), but for the analysed periods we did not find a good agreement between them, with the model underestimating the fluxes. This might be due to our analysis being focussed on the periods with low wind speeds (even though we did take that into account, and also tried using the median k value reported in Heiskanen et al. for low wind conditions). Given the poor agreement between model and data, we decided not to use the model to fill the gaps, but resort to average $F_a$ values.

Page 6 line 28: I'm assuming $Q_{10}$ is set to 2, but the authors should be explicit to make this clear.
We agree.

The manuscript now reads: "$Q_{10}$ is a non-dimensional temperature coefficient whose generally accepted value (and the value we used) for freshwater communities is 2; in the literature, values between 1.88 and 2.19 are reported", P7 L7-8 in both versions of the manuscript.

Page 7: The results section is not very well organized and is a combination of results that do seem to fit in the same paragraph. For example, the first paragraph covers results from Figure 2-5 and table 1 and does not flow well together. Break these up into individual paragraphs with topic sentences so that the reader knows what point the authors are trying to make with the paragraph.
We agree.

The assessment section is now further divided into four subsections, and some of the subsections are further divided into paragraphs. The Discussion section is also now divided into two subsections.

Page 7 line 6: do not use colloquial language such as "Anyhow, . . ."
Changed to "However, …", P7 L24 in both versions of the manuscript.

Page 7 line 12-13: remove "Figure 3 displays the NEP versus PAR."
Removed, P8 L3 in the marked-up version of the manuscript.

Page 7 lines 18-19: get rid of "In case of . . .. it was not necessary." This doesn't add anything to the fact that there was no photoinhibition.
Removed, P8 L10-11 in the marked-up version of the manuscript.

Page 7 line 20-21: why is respiration more negative with hotter years? Be explicit. Because Rh is more temperature dependent than GPP?
Yes.

The manuscript now reads: "Year 2014 was particularly hot, so the strongly negative NEP can be due to increased respiration rates, given the strong dependency of $R_h$ on temperature", P8 L10-11 in the new version of the manuscript, P8 L11-13 in the marked-up version.

Page 7 line 26: change Figg. to Figure
We removed it altogether, P8 L20 in the marked-up version of the manuscript.

Page 7 line 31-32: Get rid of "An in-depth analysis. . . some comments are possible." Since you do spend three paragraphs of the results discussing these parameters. Replace with a more constructive topic sentence.
We agree.

The manuscript now reads: "We then focused on the inter-annual variability of the values of the model parameters (reported in Table 1).", P8 L24 in the new version of the manuscript, P8 L26 in the marked-up version.

Page 8 line 7-10. Move to methods rather than results
Moved, P7 L16-19 in both versions of the manuscript.

Page 8 line 13-14: get rid of "In general, we can say that there are statistically significant differences between the years."
Removed, P9 L13-14 in the marked-up version of the manuscript.

Page 9 line11: but see Lovett et al. 2006 where NEP does not equal –NEE.
We specified it and added the reference, P10 L22-24 in the new version of the manuscript, P11 L5-7 in the marked-up version.

Page 9 line 21-22: "This is in agreement with. . ." who? The citations in parenthesis? And what is in agreement with them? That lakes are heterotrophic when NEP is negative? Or that many lakes are heterotrophic?
We rephrased and also moved the sentence to make it clearer. The references were supporting the fact that many lakes at higher latitudes are supersaturated with respect to $CO_2$, as is our lake.

The manuscript now reads, where the general results are first commented: "the net productivity values are almost always negative, meaning that the ecosystem, overall, is heterotrophic and a source of $CO_2$. In fact, the daytime and nighttime average values of the $CO_2$ flux were also always positive, albeit having lower values during the day than during the night. This is not surprising: many lakes, especially at high latitudes, are supersaturated with respect to $CO_2$ (Cole et al., 1994; Sobek et al., 2003); as a result, the $CO_2$ flux is from the lake to the atmosphere also during the day, when the aquatic primary producers are photosynthesising and absorbing $CO_2$.", P7 L24-28 in both versions of the manuscript.

Page 10 line 10-16: This is a confusing paragraph. The authors state that 1) more info on algal communities was needed to explain differences in fitted parameters, but 2) this isn't needed because the whole point of this method is to be simple and parsimonious, but 3) this method should be applied in many lakes to make links between parameters and environmental conditions / algal communities. I don't know what point the authors are trying to make with this paragraph.
We rephrased the paragraph, we hope our point is now clearer.

The manuscript now reads: "In this study, we could not clearly link the environmental variables to the changes in the Michaelis-Menten model parameters, and more information on the algal communities living in the lake would have been required in order to expand the analysis. However, it is important to stress that the simplicity of this method lies in the fact that to estimate the parameters, which can then be used to calculate the productivity, information on the algal communities is not needed. It is needed only when widening the scope of the productivity studies: when, for example, the parameters themselves and their relationship with the environmental conditions or the specific phytoplankton communities are investigated. Knowledge on the algal communities would also help when extending the productivity calculation to the whole year.", P11 L24-30 in the new version of the manuscript, P12 L18-29 in the marked-up version.

Page 10 line 28-29: How is there a comparison between the calculated NEP and modeled NEP when you fit the model to the calculated NEP? To make a statement like this, it seems like you should be training the model on a set of the calculated NEP data and then verifying with a separate set of the calculated NEP data. Also, what do you mean calculated NEP vs. modeled NEP was compared for first time? Compared for the first time using the MM method? I know there are many other examples where predictor variables are used to model lake NEP, so the authors will have to be more specific here.

We agree. We added the out-of-sample comparison.

In the manuscript there is now a new subsection in the Results section, which reads: "The analysis we performed was based on an in-sample comparison, since our goal was to check whether our method to calculate the NEP was in agreement with the PI models typically used (Michaelis-Menten, Smith (1936) and Jassby and Platt (1976) equations). However, for the Michaelis-Menten model, we also ran an out-of-sample validation for each year, in order to further verify the correspondence between the calculated NEP and the model. For each year, we randomly selected half of the data points and used them for the fit, to calculate the model parameters. Then, for the other half of the sample, we estimated the NEP using the equation and the parameters we had obtained, and compared it to the originally calculated NEP. We both evaluated the correlation coefficient r between the two NEPs (the one calculated from the data, and the one calculated from the model trained on half of the data points, then discarded), and the RMSE of the validations. The results are reported in Table 2, and show that the two NEP values compared well. The correlation coefficient r varies between 0.84 and 0.92 and the RMSE varies between 0.15 and 0.31 $\mu mol(CO_2)$ $m^{-2}$ $s^{-1}$.", P10 L8-17 in the new version of the manuscript, P10 L19-28 in the marked-up version.

The "first time" in the comparison referred to the NEP calculated with this method. It has been calculated with this method only in Hari et al. (2008), and in that paper there is no quantitative evaluation of the modelled NEP vs the calculated NEP.

We rephrased it to make it clearer, P12 L14-15 in the new version of the manuscript, P13 L15 in the marked-up version.

Page 10 lines 32-33: this is not a clear sentence.
We rephrased and expanded the sentence, we hope it is now clearer.

The manuscript now reads: "Overall, we believe that the method proposed in Hari et al. (2008) and further tested and developed here represents an improvement over the traditional approaches (bottle method and $^{14}C$ technique), given its time resolution and the fact that it is a free-water approach. We also think it is promising compared to the other more common free-water approach, the $O_2$ method, since it is direct and the respiratory quotient is not needed", P12 L18-21 in the new version of the manuscript, P13 L19-25 in the marked-up version.

I don't think figure 1 is necessary.
Removed (P18 in the marked-up version of the manuscript).

Figure 2-5: each dot represents a day or a 30 min interval? Please specify in legend
Each dot represents a 30-min interval.

The legend now states that (P17, P18, P19 and P20 in the new version of the manuscript, P18, P19, P20, and P21 of the marked-up version).

Can tables 1 & 2 be combined? It would just add 5 more columns.
Yes.

The tables are now combined (P21 in the new version of the manuscript, P22 in the marked-up version).

**RC2**

The manuscript "High-frequency productivity estimates for a lake from free-water $CO_2$ concentration measurements" presents a method to assess NEP in aquatic ecosystems. While interesting, the method requires an independent measurement of (1) the flux of $CO_2$ between the atmosphere and the water surface, (2) usage of high-frequency in situ $CO_2$ sensors, and – at least in the present approach – (3) conditions under which lateral advection fluxes (both within the water column and in the atmosphere) are limited. Overall, the methodological approach and science appear sound, but of limited utility due to known issues with eddy covariance and chamber methods for determining water to atmosphere fluxes of $CO_2$. That is, the atmospheric turbulence required for eddy flux determination is often not present at night, while the stratification required for the water column (i.e. to satisfy the assumption of no lateral fluxes in the lake) would be violated under higher wind speeds. Thus, the overall measurements are constrained to a methodological "sweet spot". The authors do not explore the potential limitations of only making measurements during these ideal conditions, which were identified for 10 summer days over a number of years for a lake in Finland. While the results for the measured days are interesting, there should be some effort to describe what these NEP rates represent relative to the other seasons (early open water after thaw, etc.), as well as efforts towards uncertainty estimates of the NEP rates and how this uncertainty cascades into the least-squares regressions for modeled parameters.

We thank the referee for the good points. We tried to follow the suggestion to describe what the NEP rates represent relative to the other seasons and to estimate the uncertainty in the NEP rates.

For the first part, the manuscript now reads: "In our case, for example, the NEP rates and hence the parameters are representative of the late summer. In lake Kuivajärvi, where diatoms are abundant, it can be expected for the productivity to have a peak in the spring and another smaller peak in the autumn, at the turnover. More measurements at those times would be needed, in order to understand whether the parameterization is still valid under those conditions.", P11 L30-33 in the new version of the manuscript, P12 L29-32 in the marked-up version.

For the second part, see the answer to the later comment about NEP uncertainty.

We would like to point out that the ideal conditions were identified in 40 days, not 10, as is specified already in the original manuscript (see P1 L9, P3 L6, P5 L22, P9 L6, P12 L11 in the new version of the manuscript, P1 L9, P3 L6, P5 L22, P9 L19, P13 L12, in the marked-up version).

The authors present in essence a case study of implementation of a method presented by Hari et al (2008), with the suggestion that the method has been overlooked and has not been used for NEP because of limited testing (P 2 L35 – P3 L1). This logic seems a bit circular, and misses the point that there are few eddy covariance studies over aquatic systems. The authors suggest that determination of the atmospheric flux of $CO_2$ could be made with chambers rather than by eddy flux, but do not discuss limitations of chamber measurements, which are not insignificant. Some discussion on how chamber measurements and in situ measurements could be co-located would be useful. As well, the study makes the assumption that $CO_2$ is uniform in the mixed layer, but this assumption does not appear to have been tested for confirmation.

We added a sentence about the potential benefit of co-locating chamber measurements and in situ measurement. However, we would like to stress that this manuscript is focused on a direct way to measure the $CO_2$ concentration in the water and on the equations used to calculate the NEP from these measurements. The flux between the lake and the atmosphere is needed in order to close the mass balance, but the methodology used to measure it (and hence a comparison of the possible methods) is beyond the scope of this paper. We tried to make this clearer in the manuscript.

The manuscript now reads: "The calculations could be improved with a better EC data set. Different methods could also be adopted to estimate the flux between the lake and the atmosphere. Chamber measurements could be used, but the time resolution could be an issue. They would need to be performed regularly. They could, however, be used to integrate the EC data set for example. Surface renewal models could also be used (e.g. Heiskanen et al. (2014)). For further information on the comparison between different flux measurement methods, see Erkkilä et al. (2018).", P11 L18-22 in the new version of the manuscript, P12 L12-16 in the marked-up version.

Regarding the uniformity of $CO_2$ in the mixed layer, for years 2010 and 2011 we had a second $CO_2$ probe at 0.5 m, whose readings matched the ones from the probe at 0.2 m.

We added this information to the manuscript: "For years 2010 and 2011, another $CO_2$ probe was located at a depth of 0.5 m, and its readings were consistent with those from the probe at 0.2 m, hence showing homogeneous $CO_2$ concentrations in the mixed layer.", P5 L11-13 in the new version of the manuscript, P5 L12-13 in the marked-up version.

It seems problematic that the authors calculate NEP based on time varying dC/dt, but use mean values for daytime and nighttime fluxes of the atmospheric flux (essentially static values). Perhaps there is additional information in the eddy flux data that could be used to propagate uncertainty in NEP calculations? For example, the standard deviation of the $F_a$ term for each day could be useful. We agree. We calculated the uncertainty on the average values of $F_a$. We decided not to use the standard deviation, since the 30-min EC data are characterised, as often happens, by large scatter. Instead, we recalculated the averages randomly selecting only half of the available data, and then we repeated the process 100 times. We then checked how far apart the calculated average values were.

We added this to the manuscript, which now reads: "We also estimated the uncertainties on the daytime and nighttime average values of $F_a$. We decided not to use the standard deviation, since individual 30-min data EC data are characterised by significant scatter. Instead, we recalculated the daytime and nighttime averages randomly choosing only half of the data in the sample, and repeated the process 100 times. Then we checked how far apart the minimum and maximum average values we obtained were, and used that as uncertainty.", P6 L28-31 in the new version of the manuscript, P6 L30-34 in the marked-up version.

The authors state (P6 L13) that the $CO_2$ flux is expected to have similar daily cycles across the analysed days, but it is not clear that the magnitude of the fluxes should be similar across days. What is the basis for this assumption?
We agree that it was not written clearly in the manuscript. It is not an assumption, but an observation, from analysing the available EC data for the studied years, and the data from years with more complete EC data sets.

We rephrased it in the manuscript, which now reads: "Under these circumstances (i.e. warm and sunny summer days without strong wind events), the $CO_2$ flux is expected to have similar daily cycles across the studied days, as is shown by the available EC data and by the EC data from years with more complete data sets.", P6 L20-22 in the new version of the manuscript, P6 L22-24 in the marked-up version.

**Specific comments**
P1 L11: Here, the model fit is described as "excellent", while later it is described as "very good" on P7 L28. Providing some metrics that would qualify as excellent should be included in the abstract.
We agree.

We changed "excellent" to very good in the abstract, and provided some metrics ($R^2 \geq 0.71$), P1 L11 in both versions of the manuscript.

P1 L19: change to ". . .in gaseous form (primarily as $CO_2$)."
We changed it, P1 L19 in both versions of the manuscript.

P4 L5: What is the permeability of silicone to $CO_2$ relative to the diffusion rate of $CO_2$ in water at the temperatures experienced in this study?
Laboratory tests on the same set-up were run for the original paper (Hari et al. (2008)). When the silicone tube was transferred rapidly from a water bath with low $CO_2$ concentration to one enriched in $CO_2$, the response time of the whole system was < 5 min (Hari et al., 2008).

P6 L29: It would be helpful to present more information describing how $p_{max}$, b and $r_0$ are determined. Which equations were used to solved for these three unknowns?
The parameter values are obtained fitting the model to the data.

We added this sentence to the manuscript, which now reads: "their values can be obtained fitting the model to the data.", P7 L12 in both versions of the manuscript.

Also see, already in the original version of the manuscript, "After calculating the NEP, we plotted the NEP versus irradiance curves. We then fitted the model (Eq. (5)) to the NEP data with the least-squares fitting method, in order to check the agreement between the data and the model and in order to estimate $p_{max}$, b and $r_0$.", P7 L13-15 in both versions of the manuscript.

P7 L26: "The curves in Figg. 2-5 have the expected trends, and this confirms that. . ." – this sounds like confirmation bias.
"confirms" was changed to "suggests", P8 L18 in the new version of the manuscript, P8 L20 in the marked-up version.

P7 L29: "This clearly indicates that the method used here allows the NEP to be accurately parameterized as a function of irradiance and water temperature." What seems to be missing here is uncertainty assessment on NEP. If NEP is not well constrained (since it is calculated from Eqn 3 assuming static rates for the daytime and nighttime $CO_2$ fluxes between the lake and the atmosphere), how can the model fits be characterized without consideration of the uncertainty in the "measured" NEP vs. the modelled fit?
We agree. We removed "accurately" (P8 L24 in the marked-up version of the manuscript), and we addressed this issue in the Discussion session.

The manuscript reads: "Our analysis was hindered by issues in the EC data set: due to inherent EC limitations and technical problems, the data set had many gaps and average daytime and nighttime $F_a$ values had to be used. The relative uncertainty on them was, on average, 50%. This uncertainty propagates to NEP through Eq. (3), and therefore to the parameter values as well. However, it does not undermine the good agreement between the model and the data, given that the average $F_a$ values were calculated putting together all the periods of the same year. Therefore, each NEP data point has the same uncertainty and the same weight in the fit.", P11 L13-18 in the new version of the manuscript, P12 L2-12 in the marked-up version.

P8 L10: "The value of b does not change significantly between any of the years" – but Table 1 and 2 show it to vary by 50% between years. This seems rather significant. The later statement that $p_{max}$ and $r_0$ are more sensitive to variation seems to be a statement that wasn't formally tested through sensitivity analysis.
We performed a statistical test to check whether the variations in the parameter values between the

years were statistically significant. See P8 L7-10 in the original manuscript, P7 L16-19 in the new and marked-up versions. The changes in the value of b are indeed large but so is the uncertainty on the value of b itself, which makes these changes not statistically significant.

P9 L14: "We hope in the future to further develop the method" – this kind of statement doesn't belong in a Discussion section.
We removed it, P11 L10-11 in the marked-up version of the manuscript.

Compare P9 L6-7: "the changes of PAR and water temperature cannot fully account for the changes in the model parameters" with P7 L29 "the method used here allows the NEP to be accurately parameterized as a function of irradiance and water temperature."
The method does allow the NEP to be parameterized as a function of PAR and water T, through the calculation of the model parameters. The model parameters change between the years, and their changes are not fully explained solely by the changes in water T and PAR, indicating that they depend on other variables as well, such as the algal community composition for example. We rephrased it to make it clearer.

The sentence from P7 L29 in the original version of the manuscript now reads: "From what is said so far, the changes of PAR and water temperature alone cannot fully account for the changes in the model parameters. The long-term variations of the parameters probably have other drivers too, such as the composition of the algal communities", P9 L28-30 in the new version, P10 L5-7 in the marked-up version.

P11 L7: I would not describe eddy covariance over a water surface as "relatively inexpensive".
"relatively inexpensive" refers to the $CO_2$ probes. The next sentence in fact reads: "The method requires at least a concomitant estimation of the $CO_2$ flux from the lake to the atmosphere. In our case the EC technique was used, which is expensive and can be laborious in the data processing phase. However, chamber measurements or surface renewal models could be equally good options.".

We tried to make it clearer in the manuscript, P12 L27-30 in the new version of the manuscript, P13 L31-P14 L1 in the marked-up version.

P17 Fig 3 for 2014: What explains the large separation between NEP values for low values of PAR and the jump in NEP just as PAR increases a bit?
The large separation is explained by choosing a PAR threshold between "night" and "day", and then having different $F_a$ average values for night and day.

We made it clearer in the manuscript, which now reads: "Note that both in Fig. 3 and Fig. 4 and especially for years 2010 and 2014 there is a large separation between NEP across the chosen PAR threshold between night and day. This is caused by having to resort to daytime and nighttime average values for $F_a$.", P8 L15-17 in the new version of the manuscript, P8 L16-18 in the marked-up version.

[revised manuscript text omitted]